# What Can Urban Buses Learn About Sunshine Regulation Adopted in Other Sectors?

Gabriel Stumpf Duarte de Carvalho * and Rui Cunha Marques 

Department of Civil Engineering, Architecture and Georesources, Instituto Superior Técnico,
University of Lisbon, Avenida Rovisco Pais, 1049-001 Lisbon, Portugal; rui.marques@tecnico.ulisboa.pt
* Correspondence: gabrielstumpf@tecnico.ulisboa.pt

**Abstract:** Since 2014, the Portuguese bus market has been experiencing substantial competition and regulatory environment changes. Nevertheless, further improvements in governance and particularly transparency are still needed. This paper intends to explore how the health and water sectors currently apply sunshine regulation to promote healthy competition and improve service quality and how good practices can be replicated into the Portuguese bus sector. We explore the strategies and tools used by the regulators from these two sectors and propose a simplified template to integrate sunshine regulation in public bus operations. The first steps for applying sunshine regulation were taken by recently enacted laws and regulations that impose the need to monitor contracts using performance indicators or other metrics. With this paper, we intend to take a further step suggesting a framework that uses these performance metrics to increase transparency and accountability in the sector.

**Keywords:** bus sector; sunshine regulation; performance indicators; sustainability

## 1. The Need to Measure the Quality of Service in the Urban Bus Sector

Public transport has a remarkable role in promoting accessibility in cities and urban areas. Hence, it is an essential tool to tackle social inequalities. According to Pereira et al. [1], from a justice perspective, the definition of accessibility is "the ease with which citizens can reach places and opportunities from a given location". Affordable transport helps the urban poor as it offers a low-cost way of reaching job opportunities, leisure activities, social services, etc. [2]. As with every public service, the crucial role of urban bus systems is to render adequate, fair, and inclusive services for the population. Thus, civic engagement is paramount to urban and transport planning, as public opinion is an essential input for the planning process of lines, schedules, and fare policy [3].

In addition, as noted by Ferreira da Cruz and Marques [4], "public transport services are considered as services of general economic interest due to their particular importance for the citizens and the society, so they should not be produced without public authorities' intervention". Consequently, regulation plays an essential role in pursuing effective competition for the market, lifting entry barriers, and guaranteeing minimum safety and quality standards [5]. The typical transport issues raised by the population relate to availability, cost, quality, and safety of public transport [2].

Quality may have several dimensions in the urban bus service, for instance: (1) Comfort: well-designed and maintained vehicle interior. In other words, availability of comfortable and clean seats, air conditioning on-board wi-fi, low-floor entrance, and so on. (2) Reliability: the buses must start and finish their service on time, and the schedules must be respected. (3) Predictability: services must be predictable, and information on timetables must be easily available to the public. (4) Safety: buses must be well maintained, and mechanical failures must be avoided and steadily corrected when they occur (5) inclusiveness: boarding equipment and exclusive areas for people with reduced mobility.

Nonetheless, some dimensions of quality can be tricky to numerically measure as they have subjective judgment. For example, cleanness can be assessed differently by two people, as one can be more tolerant than the other. However, delays and mechanical failures, for instance, are concepts well established and are common sense. These kinds of "quality dimensions" are commonly understood by all stakeholders and can be easily evaluated through performance indicators and monitored with the usage of technological tools [6].

Service quality in the form of performance indicators can be used to increase awareness regarding service provision and public budget spending. Following Nunes et al. [7], the act of public displaying performance indicators (also known as sunshine regulation) improves accountability and transparency. In that sense, the population can evaluate whether the service has been provided correctly and how the public budget has been spent. That is, whether the service is compatible/aligned with the money society spent through taxes. Since 1970, with the advent of New Public Management, governments worldwide have become aware of excessive and inefficient public spending and the need for a more "businesslike" approach to public services. Concerns about efficiency in the public realm have increased interest in measuring the performance of public services. [8–10].

Moreover, public service providers (public companies and regulated private companies) are expected to create public value (sound benefits for society) in line with environmental, economic, and social sustainability principles. Therefore, sunshine regulation can act as a tool to foster social sustainability since it can provide non-financial disclosure to meet stakeholder's expectations and to ensure adequate levels of accountability [11].

This paper intends to explore how different sectors (health and water) apply sunshine regulation and how good practices can be replicated into the Portuguese urban bus sector. Although this paper is focused on the Portuguese case, we understand that it can also inspire other places to adopt sunshine regulation in their urban bus systems. To conduct the literature review, we mainly searched the SCOPUS and Google Scholar electronic databases. In this study, we only considered papers that were published in English and were peer reviewed. Paper selection was performed using the snowballing method without specifying a time frame. We understand that experiences from past decades can be helpful to enrich the analysis and inspire new solutions.

After this brief introduction, the next section sheds light on the current state of the urban bus sector in Portugal, regarding regulation and performance indicators. Section 3 introduces the definition of sunshine regulation and presents good practices that are already being applied in the health and water sectors. Section 4 presents the possibilities and challenges of applying good practices found in the previous section in the Portuguese urban bus sector. Finally, Section 5 provides the concluding remarks of this research.

## 2. The Portuguese Urban Bus Market in a Nutshell

Public transport services are considered services of general economic interest within the European Union (EU). Due to their importance to citizens and society, competent public authorities should oversee them and intervene when necessary, guaranteeing the public interest [4].

In Portugal, the regulator for the transport sector, the Mobility, and Transport Authority (AMT in the Portuguese acronym), was set up in 2014, as a result of the institutional reorganization of the sector. Previously, this sector was regulated by the Institute for Mobility and Land Transport (IMTT) which, besides the regulatory activity, supervising road and rail transport, was responsible for the approval, licensing, and inspection of vehicles, as well as other activities such as issuing driving licenses, licensing driving schools, and training professional drivers.

Ferreira da Cruz and Marques [4] stated that the IMTT was a weak regulator due to the constitutional principle of local autonomy. That is, the central state could not interfere and did not have any sort of economic power over local governments' competencies. Moreover,

the IMTT lacked resources whereas local decision-makers lacked the know-how needed to design, procure, and monitor complex transport systems.

This restructuring made the AMT entirely focused on regulating the transport sector, including urban mobility, land, inland waterways, and rail transport. According to its statute, the AMT is responsible for supervising the procurement of public road passenger transport services, the development of general rules and principles for public transport tariff policy, supervising operators' compliance with the obligations and the proper operations, and the defense of users' rights and interests, among other functions.

Another advance in the Portuguese transport sector was the enactment of Law 52/2015, which approved the Legal Regime of Public Passenger Transport Service, in light of the European Regulation no. 1370/2007. In addition to other measures, this law establishes that the urban public passenger transport service can be only operated in two ways: directly by the competent transport authorities, using their means or by assignment, through the competitive award of a public service contract. Furthermore, the Law granted the competence to municipalities and inter-municipal communities (group of municipalities) to be transport authorities, decentralizing the central government's power. Currently, they have the power and the obligation to plan the networks and provide transport services, either by their means or by procuring through competitive tenders.

Before Law 52/2015, the governance model, the types of contracts established between authorities and operators, and even the market structure of the urban transport sector in Portugal were similar to the French case [4]. In fact, most of the problems of urban transport identified in France were also applicable to Portugal, e.g., the excessive discretionary power of the authorities during procurement procedures, insufficient competition for the market, and little transfer of risk to operators, among other shortcomings [12].

According to ANTROP [13], the National Association of Bus Transport, bus operators had to ask for authorization (in form of concession) from the IMTT to operate a line (or a group of lines) in a specific territory at their own risk. That is, the revenue (and operational) risk was fully borne by the operator. The IMTT only regulated tariff prices. The initiative for operating routes (concessions) thus belonged to the operators who defined their transport networks.

Operators had exclusive rights in their operation area and enjoyed the right of preference when a new application for a concession was submitted. This right was mainly based on the route length (km) under concession. The operator with more kilometers under concession had preference over the others to obtain the new concession. Thus, when the sector was liberalized in the 1990s, new operators had great difficulty (it was almost impossible) in granting concessions because almost the entire territory was already under concession. Moreover, if an outsider operator would intend to operate the kilometers that were not under concession, the local operator could exercise his/her right of preference and maintain dominance [13].

On the one hand, local operators had a quasi-monopoly operation maintaining a close relationship with local authorities. If a new service (or an adaptation of an existing one) was required, the local authority would negotiate with the local operator to change the service. If the adaptation were beneficial to the operator, it would accept the new service. If not, the operator would be free to deny the request. The same was true for a line that became unprofitable. The operator had the right to reduce (or even suppress) unprofitable lines. In that manner, it was common for operators to assume the role of planning the system. On the other hand, the IMTT had the power to regulate tariffs and impose discounts for specific tariffs (e.g., students). The operator had to bear the discounts and had a cap for the tariffs without any government payment or subsidy.

In 2010, there were 122 urban transport companies in Portugal, most of which were small regional operators [14]. In the cities of Lisbon and Oporto (the most important cities in Portugal), urban transport services were (and still are) operated by publicly owned companies (for instance, the case of CARRIS in Lisbon owned by the municipality and STCP in Oporto owned by a set of municipalities). At that time, the bus companies in Portugal

were remarkably inefficient due to a lack of financial resources to invest in technological improvements. Barros and Peypoch [14] identified that the Portuguese bus industry's productivity growth was small in terms of technical and technological improvements. Furthermore, some companies displayed improvements in technological change but not in technical efficiency, while other companies displayed regression in both categories. Publicly and privately owned bus companies had similar levels of productivity.

However, according to ANTROP [13], the public companies, due to the support they received from the government, had greater financial capacity to invest in better and newer fleet than their private-sector counterparts. To renew their fleet, private companies commonly had to acquire second-hand buses abroad. As public companies received substantial financial support from the government, this undoubtedly distorted the competition conditions endorsed by European Community rules. In this unbalanced scenario, private operators found it extremely difficult to invest in their fleets. In many cases, second-hand vehicles were purchased, allowing replacing a "very old fleet" with a "less old fleet".

In order to increase productivity, Barros and Peypoch [14] suggested that companies should be merged to increase economies of scale, regulation should be used as a manner to encourage bus companies to increase their efficiency, and contracts should use a yardstick strategy. Indeed, what has happened in the last decade followed this path. A series of merging and acquisitions has been happening, and concession contracts have adopted performance indicators to monitor the service provided. In 2018, the AMT [15] released a report suggesting a series of indicators for transport authorities that should be considered in the contracts as a tool to monitor them. Additionally, due to the magnitude of some contracts (e.g., metropolitan areas), different operators joined in SPVs (special purpose vehicle companies) to frequently operate together.

Although it seems that the private operators are becoming more efficient than a decade ago, especially in the metropolitan areas, ANTROP [13] states that in the rural areas, operators still need to rely on used buses bought in secondhand markets in other European countries. Since 2015, when the public tenders started to be launched, some ended empty, without any proposal, mainly in the countryside (with less attractive operations). This fact is due to a mismatch between costs (investment and operational) and expected revenues (plus insufficient subsidies estimated by transport authorities).

In sum, the last years have been busy with solid changes in Portugal's urban transport sector, mainly regarding urban buses. The results of Law 52/2015 still need time to appear and prove their value (in terms of service quality improvement). Currently, contracts are being launched in a "first-generation" format; that is, the main objective is to gather information about the operations (revenues and costs involved) and allow the transport authorities to build technical capabilities without compromising the continuity of the services to the population. In most cases, the term of these contracts ranges from four to seven years. Therefore, as expected, this first round of contracts does not provide deep network changes/optimizations nor heavy fleet investments.

The "second-generation" contracts are expected to have longer terms and bring deeper network restructurings and fleet renewal programs. In addition, in the second round, transport authorities are likely to have better information about the systems and more capacity to plan the networks and monitor and enforce public service contracts.

## 3. Sunshine Regulation

In a nutshell, sunshine regulation implies the public display of performance indicators and regular comparison (benchmarking) among service providers from the same sector [16]. Benchmarking refers essentially to the application of comparative and quantitative methods of evaluation and performance measurement of service providers over time, which allows the regulator to take decisions in the regulatory process [17]. Sunshine regulation works as an ex-post enforcement of the whole regulation [18].

According to Marques and Simões [19], the awareness of service providers' performance is fulfilled by pressure exerted over them from the media, the public through their

representative groups and non-governmental organizations, and the politicians (Government and political parties). Service providers with poor performance become "embarrassed" and will try to solve their deficiencies.

In accordance with Bolognesi and Pflieger [18], sunshine regulation requires transparency over the service provision, mainly through the reporting of performance indicators. The regulator (the principal) aims at identifying and disclosing bad practices to encourage operators (the agent) to deliver high-quality service. Consequently, it is usually called "naming and shaming" regulation because service providers fear a detrimental reputational effect. This method does not set penalties, and its coercive power is limited [19]. However, positive effects are obtained with the public display and discussion of the regulated firms' behaviour. It introduces competitiveness between them and leads to a progressive increase in performance [19]. Diverse empirical studies confirm a positive impact of sunshine regulation in diverse sectors including health, education, water, and waste [18]. Finally, sunshine regulation can be applied alone or as a complement to other regulatory measures [17].

Nevertheless, the success of the implementation of sunshine regulation is constrained by the disclosure bias problem. Bolognesi and Pflieger [18] explain that disclosure bias is poor reporting of performance indicators, and little attention is paid to practices and methodological developments to curb this issue. The authors investigated the presence of disclosure bias in the French water sector and identified that performance disclosed by the agents (53 service providers), in most cases, was different from the observed (and real) performance. They concluded that opportunistic behavior leads to disclosure biases. Cream-skimming leads to nonreporting disclosure biases, while overestimation and free-riding correspond to distortion disclosure biases. The former behavior is caused by agents disclosing performance indicators that are good to their public image and do not reveal the other indicators that can harm them [20]. The latter is based on institutional inconsistencies that could generate incentives to adopt opportunistic behavior. These inconsistencies make the gains from cheating (revealing better performance than the reality) higher than the benefits and costs of adhering to sunshine regulation [21].

In the case of the French water sector, Bolognesi and Pflieger [18] identified some causes that could have resulted in disclosure bias in sunshine regulation and pointed out some solutions:

1.  Reporting was not legally compulsory, and operators were free to report their data to the regulator. Therefore, the reporting process and the organizational structure of performance management are critical to a solid sunshine regulation.
2.  Performance indicators disclosure was significantly subject to biases which confirmed the need for direct observation (collection) of data when evaluating sunshine regulation.
3.  The usage of the same set of indicators, whatever the service structures, can lead to a misleading performance assessment.
4.  Only a "stick" approach, e.g, introducing a higher penalty for non-compliance, might lead to governance failure and opportunistic behavior. Therefore, the authors suggest that regulators should emphasize the positive impact of reporting and support a self-assessment report performed by the service providers instead of a performance evaluation conducted only by the regulator. This approach helps to create a learning supportive environment.

### 3.1. Good Practices in the Healthcare Sector

As in other sectors, regulation aims to correct the market failures in the healthcare sector. In broad terms, externalities, information asymmetry, service scarcity, market uncertainty, and monopoly/oligopoly structures are examples of market failures [22]. Healthcare regulation also intends to safeguard the basic rights of the citizens, for instance, prevent the practice of cream-skimming or even the induced demand of healthcare, leading inevitably to overtreatment [7]. In the healthcare sector, cream-skimming refers to picking patients for some characteristic(s) other than their need for care. This behavior can enhance the reputation of the provider and increase profits [23]. Under capitation or other fixed

payment schemes (such as health insurance plans), this often means picking healthier patients over others.

Healthcare regulation is taking place in many countries and for different reasons. Economic, social, political, and organizational causes paved the way for stronger healthcare regulation [7]. An example of this evolution was the introduction, in 2001 by the Department of Health in England, of an annual system of publishing 'star ratings' for public health care organizations. This gave each unit a single summary score from about 50 kinds of targets: a small set of 'key targets' and a broader set of indicators in a 'balanced scorecard' [24].

A further step was taken in April 2009 to create the British Care Quality Commission (CQC). According to the British Government, the CQC regulates all health and social care services in England, including the National Health Service (NHS). The commission ensures the quality and safety of care in hospitals and other health facilities.

The CQC publicly publishes information about the quality of individual services, including reports and ratings, to help the population choose their care. After each inspection, they produce a report and publish it on their website. The reports set out the CQC findings on each of the five essential questions targeted for the people who use the services. It describes the good practices, as well as any concerns identified. In most cases, the reports include star ratings to help citizens understand how good each local service is (Figure 1).

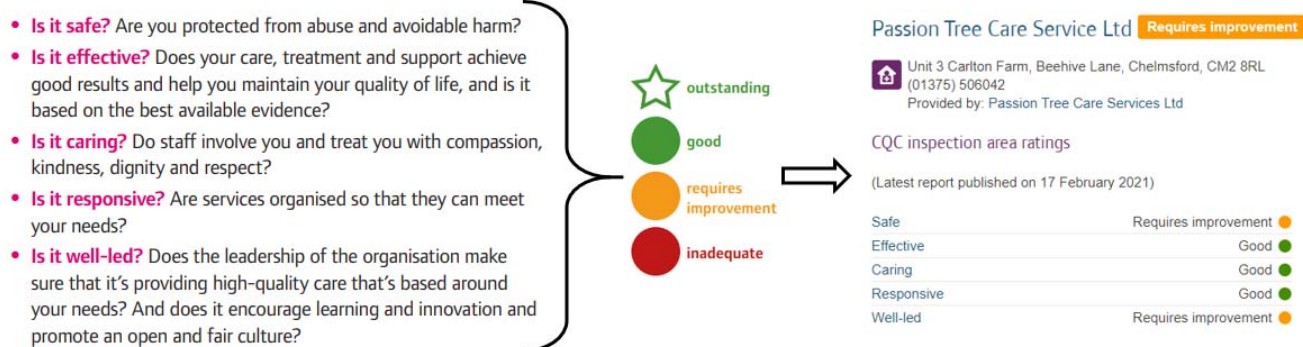

**Figure 1.** CQC rating scheme. Source: Adaptation from cqc.org.uk (accessed on 27 April 2021).

Additionally, the CQC uses a wide range of indicators to evaluate the performance of NHS hospitals, including waiting times for general practitioner referral to first outpatient appointments, vacancies in medical staffing, and the percentage of patients waiting on trolleys for more than 4 h [7].

The CQC evaluates the performance of NHS trusts through several performance indicators and checks if they accomplish the targets set by the Government. The lists of performance indicators and the methodology used to calculate them are publicly displayed on the CQC's website [7]. This approach is helpful as it sets a common language to measure performance among accountant managers, clinicians, and the public [25].

Inspired by the British experience, in Portugal, the National Health Service (SNS) developed a benchmarking process among public hospitals. Public display of performance indicators started in 2013. This process aims to improve the economic–financial performance of institutions while ensuring better performance in the provision of care to users, particularly in terms of quality and access. According to SNS, the benchmarking process allows the comparison of results and provides support to understand differences in performance, evaluating the potential for improvement of each hospital in the main areas of operation. Additionally, the regular comparison of performance indicators allows identifying best-practices.

The performance indicators are divided into three main categories: (1) Production and efficiency ratios; (2) capacity used, and (3) economic and financial evaluation. All the indicators are numerical, and hospitals can be compared among them or by geographic

regions. The platform also allows comparing a selected hospital with the national or region averages and shows the indicators' evolution through time. In that sense, it is straightforward to perceive whether a hospital financial health is getting better or worse, as well as the evolution of the number of patients assisted.

### 3.2. Good Practices in the Water Sector

The employment of benchmarking in regulation has multiple potential advantages, mainly in the water and wastewater sectors. The competition in the market is more challenging to achieve as the operators generally work in a regional (local) natural monopoly structure [16]. Following De Witte and Marques [26], this sector is usually portrayed by monopolistic characteristics and by the presence of asymmetric information (adverse selection and moral hazard), which encourages rent-seeking (raising profits without raising productivity).

Even if efficiency earnings are lost due to the increase of scale and scope diseconomies, some countries block the merging and acquisition of operators (e.g., in the UK by the Competition Authority) to keep enough players to allow regular comparison (benchmarking) in the water sector [16]. According to de Witte and Saal [27], the implementation of sunshine regulation was an essential milestone in the Dutch drinking water sector reform. The authors suggest that sunshine regulation is associated with enhanced productivity in the sector and sharing productivity gains with consumers through reduced tariffs.

Carvalho and Marques [28] conducted worldwide research about regulation and the application of benchmarking and sunshine regulation in the water sector. They sent a questionnaire to 279 regulatory entities and obtained 63 responses. They found that performance indicators were predominantly adopted in the quality-of-service regulation (95% of international cases analyzed confirmed the use of performance indicators) and, frequently, the comparison carried out was related to the performance of previous years. In around 57% of the case studies, there was a comparison with other operators, approximately 20% with reference benchmarks, and 15% with operators of other countries. In some countries (nearly 25%), penalties were imposed when performance was inadequate, sometimes by changes in the tariff structure. In some cases, users were reimbursed, whereas in others the payments went directly to the government (near 12%). In other countries, operators were ranked by their performance (8% of the situations analyzed).

In Portugal, municipalities and the Central government (the state) are responsible for water supply and sewage services. The state is responsible for inter-municipal systems and the municipalities for municipal ones only [29]. The benchmarking application is the cornerstone of the regulatory system [17]. In this domain, *Entidade Reguladora dos Serviços de Águas e Resíduos* (ERSAR), the regulatory entity, monitors 20 performance indicators for each regulated activity: urban waste, water, and sewage. Annually, ERSAR publishes a report of benchmarking that measures performance indicators and establishes explanatory factors and reference values. The group of performance indicators covers several kinds of indicators assigned by three different clusters, such as:

1. Users' interest protection (five indicators): aims to measure the quality of the services provided.
2. Sustainability of the operator (six indicators): evaluates the sustainability of the operators.
3. Environmental sustainability (three indicators): aims to measure environmental sustainability.

ERSAR uses a system of balls (scores) with different colors associated with the performance highlighted (Figure 2). With respect to each performance indicator with reference values, if the operator has a good score, it will get a green ball, a bad score will correspond to a red ball, and an average score to a yellow ball. This type of 'name and shame strategy' has proven to be very powerful [19,30]. Additionally, within the annual report, the regulator recommends actions for the entities (service providers) to improve the service in areas with unsatisfactory performance.

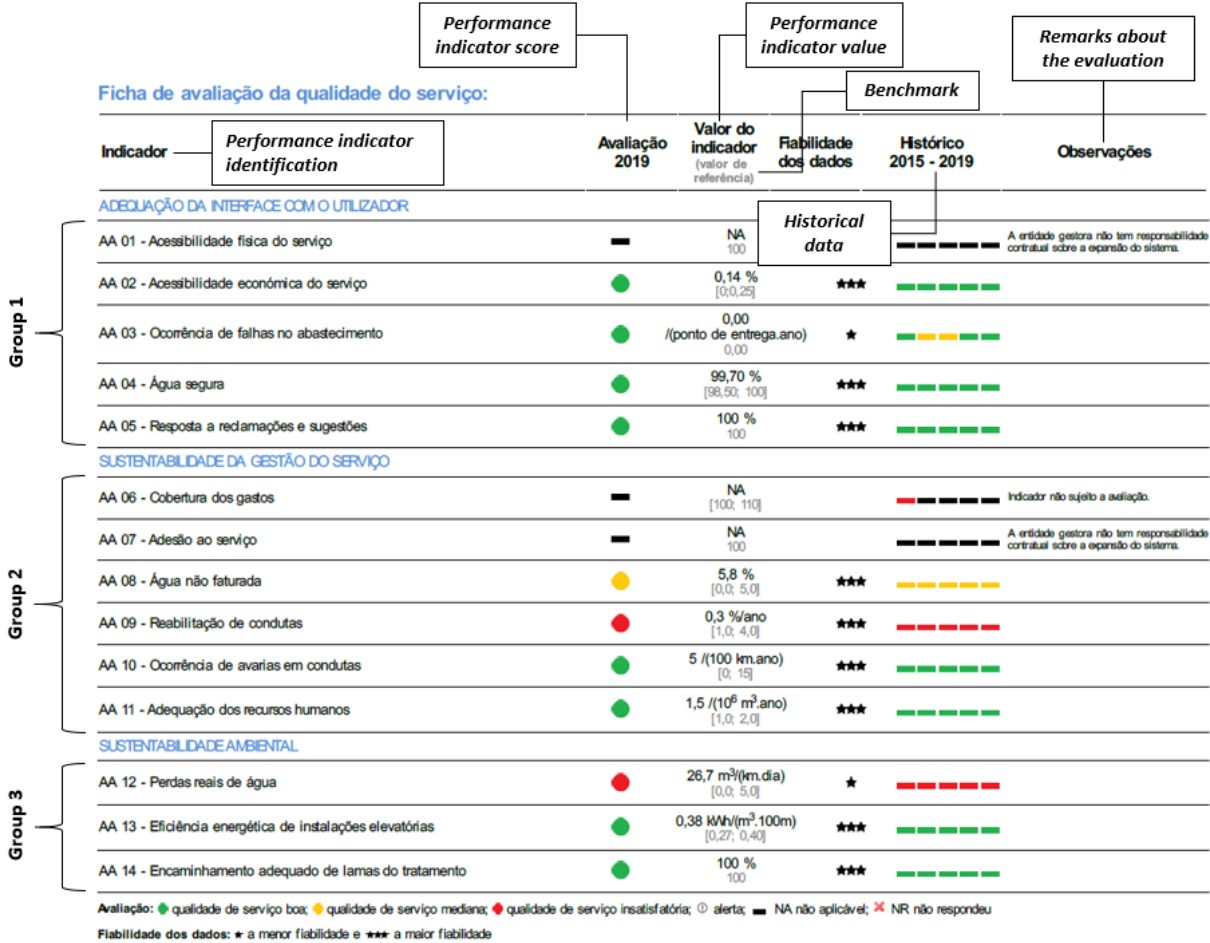

**Figure 2.** List of performance indicators monitored by ERSAR. Source: Adapted from ERSAR.

Nevertheless, in the case of Portugal, as there are concession contracts with rights and obligations well defined, the regulator is not free to make the decisions that it may consider as more adequate [19]. For instance, service providers will always have guaranteed rate of return for their operations, even if they perform poorly.

## 4. Barriers and Opportunities to Apply Sunshine Regulation in the Portuguese Urban Bus Sector

As seen in the previous sections, sunshine regulation to work properly needs a straightforward comparison among the service providers in the same sector and needs to balance the "carrot" and "stick" approach. In that sense, performance indicators must be measured and calculated based on the same methodology to enable a fair comparison. Additionally, they need to be easily understood by all the stakeholders (operators, government, and the population). To guarantee transparency, the regulator must display them periodically and publicly (at least on its website). Similar to the ones developed by SNS and ERSAR, a dashboard is a good practice that allows users to make comparisons among all the service providers and compare a specific service provider to the sector's statistics.

Regarding the bus sector, Laws 52/2015 and 18/2008 and AMT's regulations (and publications) established a building block for the sunshine regulation applications. The operators' duty of sharing performance (and financial) information allows Transport Authorities and AMT to calculate performance indicators and define benchmark values. In that sense, the AMT's report is a solid starting point. In this report, AMT [15] suggests seven groups of indicators (59 in total):

- Operational activity
- Service quality and accessibility
- Human resources
- Terminals and fleet
- Financial/economic indicators
- Information displayed to users
- Sustainability

ERSAR and NHS established rating schemes (stars rating and ball scores) that compare the service provider performance with a defined benchmark. These rating schemes are a good practice that can be easily transferred to the urban bus sector. AMT could define, for example, benchmark values for the fleet age, emissions, punctuality index, regularity index, patronage, number of complaints, mean time between failures (MTBF), mean kilometers between failures (MKBF), percentage of the fleet with AC and equipment for boarding reduced mobility people, results of the user satisfaction survey, and so on. Further, financial information about the operation (revenues and costs) can be useful to increase awareness regarding the budget and the public expenditure to maintain the service.

Nevertheless, Bolognesi and Pflieger [18] stated that applying the same group of indicators to all operations, independently of the service characteristics, is a misleading performance assessment. Therefore, only performance indicators directly related to operational performance, for example, the punctuality and regularity indexes, and MTBF and MKBF, can be used to apply the "stick" (naming-and-shaming approach). For these indicators, we suggest the use of a star rating or ball score. The relative performance can be measured according to benchmark values/targets defined by the AMT.

Other indicators related to investment capacity and capital availability are not appropriate for this use, as operators in different regions (metropolitan areas versus rural areas) can have different realities, as declared by ANTROP [13]. These kinds of indicators, such as fleet age, fleet emissions, and other features related to the technical aspects of the fleet, can be used to increase awareness regarding the regional differences of the services and guide the government to solve them. Due to that, the adoption of user satisfaction surveys for a "stick" approach needs to be taken with care. User satisfaction can be related to fleet characteristics (in most cases related to investment capacity) and operational indicators (directly related to operational performance).

A classification index is suggested to avoid the disproportionate "shaming" of small and rural area operators. Only comparable operations can be compared. It is not fair to compare metropolitan operations with rural area operations. Operations and farebox revenue are markedly different in these two scenarios. We suggest that operations can be categorized in ranges of revenue/km, revenue/passenger, passenger.km/year, and vehicles.km/year.

Finally, an online dashboard hosted on AMT's website, in the ERSAR and SNS style, would be a useful tool for the population. This dashboard could compare (in the form of ranking, for example) an operator's performance to others. This information will also allow the population (and the media) to compare private and public operators' relative performance.

## 5. A Simplified Template for Sunshine Regulation for Buses

Considering the already published material from AMT and the correlated enacted laws, Table 1 suggests a basic template to measure urban/intermunicipal service quality with a set of indicators. Additionally, we recommend that operations (consequently, the operators) should be categorized into three types: (1) Metropolitan, (2) Urban, and (3) Intermunicipal. Metropolitan operations correspond to those inside the metropolitan areas (including urban and suburban operations) of Lisbon and Oporto. Urban operations correspond to those municipal-only operations outside metropolitan areas. Finally, intermunicipal operations correspond to those among different municipalities, especially within Intermunicipal Communities.

**Table 1.** Performance indicators for sunshine regulation.

| Dimension | Sub-Dimension | Criterion | Indicator |
|---|---|---|---|
| Operational Standard | Punctuality | Services on time | Number of services on time at the first stop/total number of services |
| | Reliability I | Services completed | Number of services completed/Total number of services |
| | Reliability II | First and last services completed | % of first and last services completed |
| | Maintenance | Breaks on service | mean kilometres between failures (MKBF) |
| Comfort | Connectivity | Availability of wi-fi onboard | % of fleet with free wi-fi onboard |
| | Climate comfort | Availability of A/C | % of fleet with A/C |
| | Access | Boarding convenience | % of fleet with low entry |
| Inclusiveness | Accessibility | Boarding convenience | % of fleet with boarding equipment for PRM |
| Resources | Financial | Revenue | Revenue/Passenger |
| | Assets | Fleet | km/bus |
| Perceived quality | Overall service quality | User satisfaction | User satisfaction survey |

Following the example of ERSAR, AMT can define benchmark values for each performance indicator, and according to them, define a star or ball rating. Experts can set these values based on research with the current Portuguese operations or international reviews, such as the International Bus Benchmarking group. All operators can be compared based on the same benchmark values for the operational performance indicators. We understand that these indicators only depend on the operator's managerial skills and are not affected by capital and investment capacity.

If the regulator understands that it is needed, the indicators for the other four dimensions can have different benchmark values for each type of operation. As mentioned in Section 4, maybe there is a substantial difference in the investment capacity among operators from different operation categories, and these limitations can directly affect the result of performance indicators in these four dimensions.

Finally, to prevent disclosure bias, the operational performance indicators should be calculated based on data from automated vehicle locations (AVL) and subjected to independent validation [31]. Figure 3 presents an illustrative example with the corresponding responsibilities of the operators, transport authorities (TA), and the regulator.

On the one hand, operators are responsible for managing and maintaining the monitoring systems (AVL and ticketing systems). They also need to allow the TA to access the data and calculate the performance indicators. On the other hand, the TA must be responsible for applying the user satisfaction survey and conducting audits. In possession of the performance indicators, the TA needs to send them to the regulator's (AMT in Portugal) database. The regulator must gather the information from all the operations under its supervision, classify them in a star rating scheme and publicly display the results on its website in the form of a dashboard. In addition, the regulator should be responsible for the elaboration of the user satisfaction survey template to guarantee that all TA will apply the same survey and operators will be compared by the same standards.

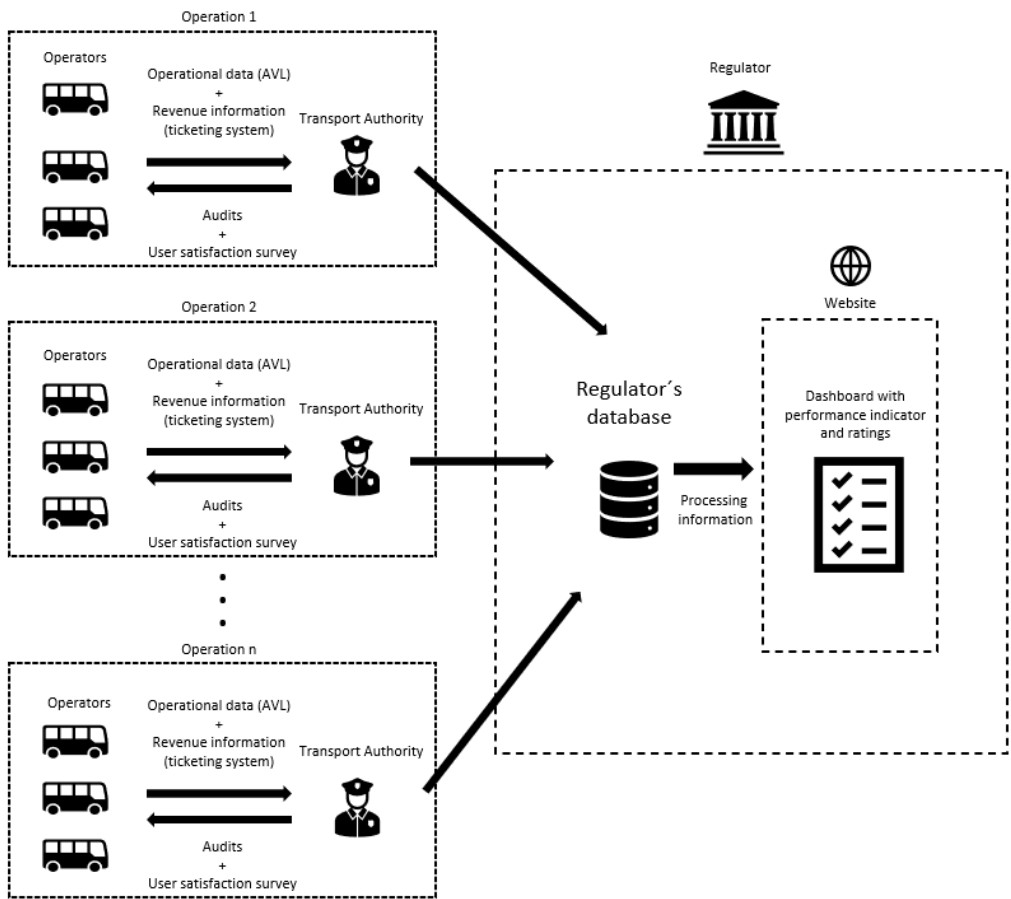

**Figure 3.** A conceptual framework for sunshine regulation in the bus sector.

## 6. Concluding Remarks

Since 2014, the Portuguese bus market has been experiencing substantial changes in terms of competition and regulatory environment. The establishment of AMT as a proper regulator separated the "Regulator" State from the "Granting" State, allowing a more efficient regulation of the bus sector in Portugal. Law 52/2015, enacted in light of the European Regulation no. 1370/2007, delimited the options of how bus public services must be delivered and operated. Additionally, this law transferred the central government's responsibility to be the granting authority of urban, metropolitan, and regional public regular services to municipalities, metropolitan areas, and inter-municipal communities. With that measure, these new Transport Authorities took the responsibility of planning the network and contracting and monitoring the services.

The current contracts must contain performance indicators, following those suggested by AMT in its publications. Although AMT already defined the requisites for monitoring the contracts, such as how performance indicators should be calculated and at which frequency, it does not use them to increase transparency in the sector and apply the sunshine regulation.

This paper investigated how the health and water sectors (cases in the UK and Portugal) are applying sunshine regulation and its subsequent results. We expect this paper to shed light on the importance of sunshine regulation and on the possible positive effects on the bus market in Portugal (and other regions worldwide).

The benchmarking process as a naming-and-shaming approach has been successfully used in these sectors to promote a healthily competition among service providers. In the case of public bus operations, a few precautions must be taken. Some performance indicators (mainly the operational ones) are directly related to the operator's management ability, while others are more dependent on financial and investment capacities,

which may significantly vary among operators from different regions and under different contractual agreements.

Due to that, we suggest that only operational performance indicators should be used as a "stick" approach using a star rating scheme. Operators with bad performance (constant delays, mechanical failures, etc.) will get a low score (one star, for example), and operators with outstanding performance will get higher scores. In contrast, indicators related to comfort, inclusiveness, resources, and perceived quality should be used to increase awareness regarding the overall quality of the service and guide public policies.

Finally, to avoid disclosure bias, all performance indicators must be calculated by the Transport Authorities using electronic systems such as the AVL and ticketing ones. In addition, they must be responsible for the audits and the user satisfaction survey.

**Author Contributions:** Conceptualization, G.S.D.d.C. and R.C.M.; methodology, G.S.D.d.C. and R.C.M.; resources, G.S.D.d.C.; data curation, G.S.D.d.C.; writing—original draft preparation, G.S.D.d.C.; writing—review and editing, R.C.M.; supervision, R.C.M. All authors have read and agreed to the published version of the manuscript.

**Funding:** This research received no external funding.

**Institutional Review Board Statement:** Not applicable.

**Informed Consent Statement:** Not applicable.

**Conflicts of Interest:** The authors declare no conflict of interest.

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
