# Peer review of "What Can Urban Buses Learn about Sunshine Regulation Adopted in Other Sectors?"

_sustainability, doi:10.3390/su13126694_

Round 1
Reviewer 1 Report
The paper ‘What urban buses can learn about sunshine regulation adopted in other sectors?’ deals with a crucial dimension for the provision of services of general economic interest: bus sector regulation based on service quality. Selecting the Portuguese experience, authors describe water and health services as possible motivations to conduct a reform for the national bus reform through a sunshine regulation. The paper is rich while describing national experience for the three selected cases. The message flows due to good writing, thus facilitating the readers understanding. A few points this reviewer would like do discuss in order to make a contribution. (i) What are theoretical and practical implications of this research piece? Authors should reinforce the practical contribution of the study in the introduction section since it makes a clear contribution to the national bus quality service regulation. (ii) On page 63, public value. The concept needs clarification especially when it comes to the discussion between bus services in the two biggest cities and other areas. In this, services quality and not efficiency or productivity became major drivers for public policy implementation and evaluation – see Pollitt and Bouckaert (2017). (iii) Specific terms for conducting your research needs to be had from 70 to 74 lines and if others find necessary to reply or develop this own study (cf. Bryman 2016). (iv) If major causes for inefficiency comes from the negotiation between Principals and Agents, the research from Marques and Silvestre (2017) must be included in this analysis since focus the regulator inefficiency action in the face of raising transaction costs between involved actors. (v) Finally, everything started with the New Public Management paradigm that had been described by Hood (1991). You should include this paper so that the reader could understand implications for regulation coming with the privatization and decentralization movement that begun in the three decades. Good luck with your research!
Author Response
Dear reviewer,
We appreciate your remarks and your suggestions. We will have a detailed look at all references that you suggested and introduce them when is the case.
Thank you very for your time spent reviewing our paper,
Gabriel Carvalho and Rui Marques
Reviewer 2 Report
The studied theme is very interesting because explore how the health and water sectors currently apply sunshine regulation to promote healthy competition and improve service quality and how good practices can be replicated into Portugal urban bus sector. The Introduction, literature review and methodology are logic, well written, prepared and explained. Perhaps adding some new papers would be more beneficial for the readers because of the insufficiently theoretical part.
Author Response
Dear reviewer,
I appreciate your remarks and your suggestions. Although sunshine regulation is a not new theme, we found some difficulty finding references, especially in the public transport sector. We will try to enrich the paper with more references.
Thank you very for your time spent reviewing our paper,
Gabriel Carvalho and Rui Marques
Reviewer 3 Report
The paper titled "What urban buses can learn about sunshine regulation adopted 2 in other sectors?" argues for the application of sunshine regulation in the Portugal bus market given positive experiences in other sectors of the economy in other countries. It would be useful if the paper added discussion about how the proposed template for sunshine regulation for buses can be evaluated in practice.
Author Response
Dear reviewer,
I appreciate your remarks and your suggestions. Although sunshine regulation is a not new theme, we found some difficulty finding references, especially in the public transport sector. We will try to enrich the paper with more references and a discussion over the practical evaluation of the proposed template.
Thank you very for your time spent reviewing our paper,
Gabriel Carvalho and Rui Marques